# Prognostic Significance of the Myelodysplastic Syndrome-Specific Comorbidity Index (MDS-CI) in Patients with Myelofibrosis: A Retrospective Study

**DOI:** 10.3390/cancers15194698

**Published:** 2023-09-24

**Authors:** Kira-Lee Koster, Nora-Medea Messerich, Thomas Volken, Sergio Cogliatti, Thomas Lehmann, Lukas Graf, Andreas Holbro, Rudolf Benz, Izadora Demmer, Wolfram Jochum, Tata Nageswara Rao, Tobias Silzle

**Affiliations:** 1Clinic for Medical Oncology and Hematology, Cantonal Hospital St. Gallen, 9007 St. Gallen, Switzerland; 2Department of Intensive Care, Cantonal Hospital St. Gallen, 9007 St. Gallen, Switzerland; 3ZHAW School of Health Sciences, Institute of Public Health, 8400 Winterthur, Switzerland; 4Institute of Pathology, Cantonal Hospital St. Gallen, 9007 St. Gallen, Switzerland; 5Centre for Laboratory Medicine, 9001 St. Gallen, Switzerland; 6Division of Hematology, University Hospital of Basel and University of Basel, 4001 Basel, Switzerland; 7Division of Hematology and Oncology, Spital Thurgau AG, 8569 Muensterlingen, Switzerland; 8Laboratory of Stem Cells and Cancer Biology, Department of Medical Oncology and Hematology, Medical Research Center, Cantonal Hospital St. Gallen, 9007 St. Gallen, Switzerland; 9Institute for Pharmacology, University of Bern, 3012 Bern, Switzerland

**Keywords:** myelofibrosis, prognostic systems, comorbidities, MDS-CI, DIPSS, MIPSS70

## Abstract

**Simple Summary:**

To assess the prognosis of myelofibrosis (MF), one takes into account age and the degree of anemia and leukocytosis together with the presence of very immature cells (“blasts”) in the peripheral blood and constitutional symptoms (fever, night sweats and weight loss). Since both disease- and patient-related factors determine the course of disease, we investigated the influence of comorbidities on the prognosis of MF. For this purpose, we applied the Myelodysplastic Syndrome-Specific Comorbidity Index (MDS-CI), which offers a comprehensive tool to assess the extent of comorbidities in a structured way. Cardiac diseases and solid tumors were the comorbidities most often observed in our cohort and overall survival showed significant differences between the single risk groups of the MDS-CI. In addition, we found that the MDS-CI provided prognostic information independently from the standard tool of prognostication, the Dynamic International Prognostic Scoring System (DIPSS), and a related score, which additionally takes the mutational profile of the disease into account (Mutation-Enhanced International Prognostic Scoring System (MIPSS)-70). Taken together, our study suggests that the MDS-CI represents a valuable tool to identify MF patients with an increased vulnerability due to comorbidities.

**Abstract:**

In myelofibrosis, comorbidities (CMs) add prognostic information independently from the Dynamic International Prognostic Scoring System (DIPSS). The Myelodysplastic Syndrome-Specific Comorbidity Index (MDS-CI) offers a simple tool for CM assessment as it is calculable after having performed a careful history and physical examination, a small routine chemistry panel (including creatinine and liver enzymes) and a limited set of functional diagnostics. To assess the prognostic impact of the MDS-CI in addition to the DIPSS and the Mutation-Enhanced International Prognostic Scoring System (MIPSS)-70, we performed a retrospective chart review of 70 MF patients who had not received allogeneic stem cell transplantation (primary MF, n = 51; secondary MF, n = 19; median follow-up, 40 months) diagnosed at our institution between 2000 and 2020. Cardiac diseases (23/70) and solid tumors (12/70) were the most common CMs observed at MF diagnosis. Overall survival (OS) was significantly influenced by the MDS-CI (median OS MDS-CI low (n = 38): 101 months; MDS-CI intermediate (n = 25): 50 months; and high (n = 7): 8 months; *p* < 0.001). The MDS-CI added prognostic information after inclusion as a categorical variable in a multivariate model together with the dichotomized DIPSS or the dichotomized MIPSS70: MDS-CI high HR 14.64 (95% CI 4.42; 48.48), *p* = 0.0002, and MDS-CI intermediate HR 1.97 (95% CI 0.96; 4.03), *p* = 0.065, and MDS-CI high HR 19.65 (95% CI 4.71; 81.95), *p* < 0.001, and MDS-CI intermediate HR 1.063 (95% CI 0.65; 4.06), *p* = 0.2961, respectively. The analysis of our small and retrospective MF cohort suggests that the MDS-CI represents a useful tool to identify MF patients with an increased vulnerability due to comorbidities. However, analyses of larger cohorts are necessary to define the value of the MDS-CI as a prognostic tool in comparison with other comorbidity indices.

## 1. Introduction 

Myelofibrosis (MF) comprises a heterogeneous group of *BCR::ABL1* egative myeloproliferative neoplasms (MPNs) [1]. They arise de novo as primary MF (PMF) or represent the advanced stage of a pre-existing MPN (secondary MF; SMF) and are characterized by genetic alterations in hematopoietic stem or progenitor cells [2,3] together with inflammatory changes that arise in the bone marrow microenvironment but systemically affect the whole organism [4,5].

To assess MF prognosis, the age, extent of anemia and leukocytosis, constitutional symptoms and presence of blasts in the peripheral blood are the basic factors considered by the (Dynamic) International Prognostic Scoring System (IPSS and DIPSS, respectively) as standard tools for prognostication [6,7]. These scores can be refined by including additional clinical factors or the genetic profile of the disease as depicted by chromosomal aberrations and/or the mutational profile [8,9].

Apart from disease-related factors, comorbidities have been shown to impact MF prognosis. This is the case for individual comorbidities [10,11,12] and for the burden of comorbidities as assessed by comprehensive indices like the Charlson Comorbidity Index (CCI), the Hematopoietic Cell Transplant Comorbidity Index (HCT-CI) or the Adult Comorbidity Evaluation-27 (ACE-27) [13,14,15,16]. 

The Myelodysplastic Syndrome (MDS)-Specific Comorbidity Index (MDS-CI) is based on the HCT-CI and consists of a set of comorbidities of four major organ systems (heart, lung, liver and kidneys) together with solid tumors that have been shown to significantly affect survival in MDS patients [17,18]. From a clinical point of view, MDS and MF share some overlapping features: cytopenias, especially anemia and (in the myelodepletive phenotype of MF [19]) even thrombocytopenia or neutropenia, are common clinical problems, as are an increase in blasts and a propensity to progress into acute myeloid leukemia. 

Therefore, we wanted to assess the prognostic potential of the MDS-CI within the context of both the DIPSS, which represents the standard tool for prognostication, and the Mutation-Enhanced International Prognostic Scoring System (MIPSS)-70, which considers the mutational profile in addition to conventional risk factors. We performed a retrospective chart review of patients diagnosed at our institution between 2000 and 2020.

## 2. Materials and Methods

### 2.1. Study Cohort

Patients diagnosed with MF at the Cantonal Hospital St. Gallen between 2000 and 2020 were included in this monocentric chart review. Patients who had received allogeneic stem cell transplantation were not included in the study cohort because the IPSS-R-independent prognostic value of the MDS-CI was established in a cohort of non-transplanted MDS patients as well [18]. We reviewed all cases on an individual basis in order to ensure a correct classification according to the WHO 2016 classification. 

### 2.2. Data Collection

We retrospectively collected clinical and laboratory data from hospital records as documented at time of diagnosis (±30 days) and before the start of treatment. For patients diagnosed at our center but treated elsewhere, we retrieved the respective information from the treating physicians. 

### 2.3. Molecular Profiling

For patients whose diagnostic work-up did not include next-generation sequencing (NGS), molecular profiling was performed using the Oncomine Myeloid Research Assay (Thermo Fisher Scientific, Waltham, MA, USA) if DNA was available from the respective samples obtained at diagnosis, as described in the Appendix A.

### 2.4. Prognostic Scoring Systems 

The DIPSS was determined as described by Passamonti et al. [7], the MIPSS70 was calculated according to Guglielmelli et al. [20] and the MDS-CI was calculated according to Della Porta et al. [17]. The variables and risk groups of the MDS-CI are shown in Appendix A.

### 2.5. Statistical Analysis

Categorical variables were analyzed using frequency tables and compared using the χ^2^ test. Continuous variables were described using the median and interquartile range (IQR) and compared using the Mann–Whitney U test (comparison of two groups) or the Kruskal–Wallis test (comparison of ≥3 groups) because the data did not follow a normal distribution. Overall survival was calculated in months from the date of diagnosis to the respective event date; i.e., death or censoring. Kaplan–Meier estimates with a log-rank test or Breslow test and Cox proportional hazard regression models with robust standard errors were employed to estimate the unadjusted and adjusted survivor functions. We reported survivor functions and hazard ratios (HR) with corresponding 95% confidence intervals (95% CI). The *p*-values for the comparison of the Kaplan–Meier curves were derived from the log-rank test, if not stated otherwise. The likelihood ratio test (LR test) was used to assess the additional prognostic value of individual parameters in an unrestricted model compared with the restricted model comprising the DIPSS or MIPSS70 only. Moreover, we reported the respective concordance index (C-index) as a metric of the model performance. The C-index has been widely applied in the context of Cox proportional hazard models and relates to the agreement between observed outcomes and predictions. All statistical tests were two-sided and a *p*-value < 0.05 was considered to be statistically significant. All analyses were performed using IBM SPSS Statistics for Windows (Version 25.0., IBM Corp, Armonk, NY, USA) or Stata (Version 17.0, StataCorp, College Station, TX, USA).

## 3. Results

### 3.1. Patient Population

In total, 70 patients (male (n = 37) and female (n = 33); median age 73 years and range 28–87) were evaluable (primary MF, n = 51; MF post-ET or -PV, n = 19). During follow-up (median 40 months; range 0–184) 35/70 (50%) patients died and 2/70 (2.9%) were lost to follow-up. The patient characteristics are shown in detail in Table 1.

The majority of patients received a cytoreductive therapy (27/70 hydroxyurea (39%), 8/70 anagrelide (11%) and 3/70 interferon (4%)). Ruxolitinib was administered to 37/70 patients (53%). Therapies less often used included recombinant erythropoietin (10/70 patients (14%)), immunomodulatory drugs (thalidomide or lenalidomide; 3/70 (4%)), steroids (2/70 (3%)) and danazol (1/70 (1.4%)). One patient (1.4%) underwent a splenectomy and three patients (4%) underwent splenic irradiation.

The patient characteristics of the seven patients within the high-risk category according to the MDS-CI are shown in Appendix A. In total, 5/7 patients received ruxolitinib. In one patient, MF-associated thrombocytopenia prevented the use of ruxolitinib because chemotherapy had to be urgently applied to treat a metastatic neuroendocrine carcinoma simultaneously diagnosed with the post-PV MF. A second patient was diagnosed in 2010 and died before the availability of the drug in Switzerland. 

In 59/70 patients (84%), a driver mutation was identified (*JAK2* V617F, 39/70 (55%); MPL, 4/70 (6%); JAK2 and MPL, 2/70 (2.8%); and CALR, 14/70 (20%)). In total, 3/70 patients (4.2%) were truly triple negative. In 8/70 patients (11.4%), no driver mutation was identified but the work-up was incomplete and no DNA was available for a retrospective analysis.

### 3.2. Prevalence of Comorbidities at Diagnosis and Hematological Parameters and Markers of Systemic Inflammation in Patients with and without Comorbidities

Comorbidities according to the MDS-CI were present in 32/70 patients (46%). The most frequent were cardiac diseases (23/70, 33%) and solid tumors (12/70, 17%). Hepatic, pulmonary and renal comorbidities were observed less frequently (4/70 (6%), 3/70 (4%) and 1/70 (1%), respectively). In total, 23/70 patients (33%) had one comorbidity; combined comorbidities were observed in 9/70 patients (13%; 7/70 had two comorbidities (10%) and 2/33 had three comorbidities (3%)).

Patients with comorbidities were significantly older than patients without (median 77 years [95% CI 67–80] versus 70 years [95% CI 59–76]; *p* = 0.005). There was no difference between male and female patients with regard to the presence of comorbidities (comorbidities were present in 17/37 (46%) male patients and in 15/33 (46%) female patients; *p* = 0.967). 

With regard to peripheral blood values, we observed no differences between MF patients with or without comorbidities. However, patients with comorbidities had significantly higher levels of CRP (median 9 mg/L versus 3 mg/L; *p* < 0.001) and ferritin (median 210 mg/L (IQR 116–396) versus 122 mg/L (IQR 55; 176); *p* = 0.009). With regard to the levels of albumin, no difference was observed (for details, see Table 1). These observations were confirmed in a separate analysis of the 51 patients with primary MF, as shown in Appendix A.

### 3.3. Impact of Comorbidities according to the MDS-CI on Survival

As shown in Figure 1, median overall survival (OS) was significantly influenced by the absence or presence of comorbidities classified according to the MDS-CI at the time-point of diagnosis. The single MDS-CI subgroups showed the following OS: low (n = 38), 101 months; intermediate (n = 25), 50 months; and high (n = 7), 8 months; *p* < 0.001.

Compared with patients without any comorbidities according to the MDS-CI (n = 38), patients with cardiac disease (n = 23) had a significantly shorter OS (median 50 months [95% CI 27–73] versus 101 months [95% CI 65–137]; *p* = 0.001). This was the case for patients with a solid tumor (n = 12) as well (median 25 months [95% CI 5–45] versus 101 months [95% CI 65–137]; *p* = 0.021) (Appendix A). 

### 3.4. Impact of the MDS-CI in the Context of the DIPSS

The DIPSS was available for all patients (low, n = 8; intermediate-1, n = 30; intermediate-2, n = 24; high, n = 8), with significant differences in the OS between the single risk groups as expected (see Appendix A).

As no deaths occurred in the DIPSS low-risk group during the observation period, we split the cohort into two risk groups for further survival analyses. The first group (“DIPSSdich^low^”, n = 38) included DIPSS low and intermediate-1 patients and the second group (“DIPSSdich^high^”, n = 32) comprised DIPSS intermediate-2- and high-risk patients (see Appendix A). 

Within the DIPSSdich^high^ group, we observed significant differences in the OS between the three MDS-CI subgroups (MDS-CI low: median 54 months; MDS-CI intermediate: median 50 months; and MDS-CI high: median 8 months; *p* < 0.001; see Figure 2). Similarly, within the DIPSSdich^low^ group, the OS of patients with a low MDS-CI was longer than the OS of patients with an intermediate MDS-CI (median 117 months [95% CI 97–137] versus 89 months [95% CI 30; 148]), but this difference failed to reach a statistical significance following a comparison using the log-rank test (*p* = 0.092). A comparison with the Breslow test yielded a significant result (*p* = 0.022). Neither of the two MDS-CI high-risk patients contained in the DIPSSdich^low^ group died during the observation period (see Figure 2A,B).

After the inclusion of the MDS-CI as a categorical variable (reference group: low-risk) in a model with a dichotomized DIPSS, the MDS-CI high-risk category retained its independent prognostic value (HR 14.64; 95% CI 4.42–48.48; *p* < 0.001), whereas for the MDS-CI intermediate-risk category, the prognostic impact was only of borderline significance (HR 1.97; 95% CI 0.96–4.03; *p* = 0.065; see Table 2). The DIPSSdich-independent prognostic value of the MDS-CI high-risk group was confirmed in a separate analysis that included only cases with primary myelofibrosis (see Appendix A).

### 3.5. Impact of the MDS-CI in the Context of the MIPSS70

The MIPSS70 was available for 56 patients (MIPSS70 low, n = 2; MIPSS70 intermediate, n = 42; MIPSS70 high, n = 12), with significant differences in OS between the single groups (see Appendix A).

As the patient number was low (n = 2) and no deaths occurred within the MIPSS70 low-risk group, we again dichotomized our cohort into a “MIPSS70dich^low/int”^ group (comprising MIPSS70 low- and intermediate-risk, n = 44) and a “MIPSS70dich^high”^ group (high-risk, n = 12) for further survival analyses (see Appendix A).

The MDS-CI separated both groups of the dichotomized MIPSS70 into three groups with significantly different OS, despite the very low patient numbers in the MIPSS70dich^high^ group (MIPSS70dich^low/int^ median: 115, 89 and 11 months; *p* = 0.006; see Figure 3) (MIPSS70dich^high^ median OS: MDS-CI low (n = 2) 54 months [95% CI 34; 74]; MDS-CI intermediate (n = 8) 25 months [95% CI 13;87]; and MDS-CI high (n = 2) one month [95% CI 0; 21]; *p* = 0.001).

In a multivariate Cox regression analysis as a categorical variable (reference group: low-risk), the MDS-CI added prognostic information after its inclusion in a multivariate model together with the dichotomized MIPSS70 (MDS-CI high-risk HR 19.65; 95% CI 4.71; 81.95; *p* < 0.001MDS-CI intermediate HR 1.63; 95% CI 0.68–1.06; *p* = 0.296; see Table 2). This was confirmed in a separate analysis that included only cases with primary myelofibrosis (see Appendix A)

### 3.6. Additional Prognostic Value of the MDS-CI and Model Performance

Likelihood ratio tests for both MIPSS70dich (*p* = 0.0017) and DIPSSdich (*p* = 0.0018) showed that adding the MDS-CI group as an additional predictor significantly increased the prognostic value of the model. This observation was further substantiated by the lower AIC and BIC values, indicating a better model fit for the respective models including the MDS-CI groups. Similarly, models including the MDS-CI groups yielded a higher concordance index, indicating that the predictor added more prognostic information to the model (see Table 3).

## 4. Discussion

Our small and retrospective study demonstrates that the applicability of the MDS-CI can surpass prognostication in MDS because it can be used for the prognostication of patients with MF as well. The MDS-CI identified MF patients as being more vulnerable due to their comorbidities, even after stratification according to either DIPSS or MIPSS70. 

Several scoring systems are available to assess the prognostic impact of comorbidities in a structured way (for a review, see [21]). To our knowledge, currently, two scores have been evaluated in the context of IPSS and DIPSS: the Adult Comorbidity Evaluation-27 (ACE-27) and the Hematopoietic Cell Transplantation Comorbidity Index (HCT-CI). Data on the prognostic utility of comorbidities within the context of prognostic scoring systems taking the molecular risk profile into account are currently lacking.

While the HCT-CI is a well-established tool to assess the mortality risk associated with allogeneic stem cell transplantation in MF [22,23], its prognostic value in a non-transplant setting seems to be limited. A high HCT-CI score (≥3) was not significantly associated with an increased risk of all-cause death in an analysis of a Canadian cohort of 306 MF patients [13]. In a cohort from Serbia (n = 131), the single risk groups of the HCT-CI were associated with significant survival differences but failed to add prognostic value to multivariate models together with the IPSS [15]. 

The ACE-27 represents an additional tool that captures a greater spectrum of cardiovascular and venous thrombotic diseases in comparison with the HCT-CI [13]. For a high ACE-27 score (≥3), an almost doubled risk of all-cause mortality has been reported [13] and, more importantly, its prognostic value has been shown to be independent from both IPSS [15] and DIPSS [16].

Our observation of a DIPSS-independent prognostic value of comorbidities as assessed by the MDS-CI confirmed these observations regarding the impact of comorbidity in the context of the DIPSS. In addition, we could show that comorbidities assessed according to the MDS-CI added prognostic information, even if one has considered the mutational profile. 

To be calculated, the MDS-CI requires only basic clinical information derived from a carefully taken history and physical examination, together with basic laboratory analyses and a limited set of easily available technical examinations (electrocardiography, echocardiography and pulmonary function tests) [17]. As it is easy to determine, it might be particularly useful in clinical routines, even if it is less comprehensive in terms of the disease spectrum that is considered.

However, the MDS-CI still covers several medical conditions of the major organ systems commonly encountered in MF patients. It takes into account comorbidities related to the heart, lungs, kidneys and liver together with the presence of solid tumors. In MF patients, a high rate of deaths due to cardiovascular events and secondary cancers has been reported [24]. The greatest non-cancer mortality in MF patients is due to heart disease [25]. Correspondingly, cardiac comorbidities were by far the most common comorbidity in our dataset. As the MDS-CI gives cardiac comorbidities a high weight, it is well-suited to capture their prognostic relevance in MF. 

Renal insufficiency is a common problem in MPN [26,27] and is associated with increased mortality [10,11,12]. The liver function may be impaired in MF directly by vascular complications or extramedullary hematopoiesis, which can lead to portal hypertension or liver cirrhosis [28,29]. If present, viral infections of the liver independently affect prognosis from the IPSS, as described for Hepatitis C [10]. In addition, pulmonary disease independently affects the prognosis of MF from the IPSS [10].

The fact that the MDS-CI focuses on this core set of comorbidities may represent an advantage. For more comprehensive scoring systems like the Comorbidity Illness Rating Scale (CIRS), it has been shown that high scores are sometimes reached by a combination of comorbidities of questionable relevance quoad vitam, especially in younger MF patients [30].

In comparison with the ACE-27, one major disadvantage of the MDS-CI, if applied to MF patients, could be that it considers neither venous thromboembolism and peripheral and cerebrovascular disease nor comorbidities representing cardiovascular risk factors like diabetes or hypertension. However, a large study from Spain involving 668 MF patients showed that several of the well-known risk factors for cardiovascular disease (diabetes, hypertension, dyslipidemia and smoking) were associated with an increased risk of death in a univariate analysis, but failed to show an additional prognostic impact in a multivariate model including the IPSS [10]. This indicates that in MF, these cardiovascular risk factors do not represent relevant comorbidities in terms of prognosis, even if their proper management remains of high importance for the management of MF patients. 

In addition, atrial fibrillation (AF) is highly weighted in the MDS-CI (giving 2 points as a marker of cardiac disease) and there is a well-established association of AF with cardiovascular risk factors like obesity, arterial hypertension, diabetes mellitus and smoking [31]. The same applies for peripheral arterial disease [32] and cerebrovascular disease [33,34]. One could, therefore, speculate that within the MDS-CI, AF serves as a surrogate marker for the presence of cardiovascular risk factors and/or subclinical cardiovascular disease. 

From a clinical point of view, our results clearly show that considering comorbidities in the care of MF patients is of great importance. Given the prominent role of vascular complications, the optimal management of risk factors and appropriate care for manifest diseases are essential. In most countries, the JAK2-inhibitor ruxolitinib is currently reimbursed only for MF patients with intermediate- or high-risk disease in order to control systemic symptoms or splenomegaly. For both the pathophysiology of MF [4,5] and comorbidities arising in MF patients [35], systemic inflammation plays a central role. The significantly higher values for CRP and ferritin as acute phase proteins in MF patients with comorbidities in our cohort probably reflect this phenomenon. The use of ruxolitinib may be of clinical benefit to MF patients with lower-risk disease as well, if relevant comorbidities are present [35], as it was shown that the use of ruxolitinib modified the negative prognostic impact of diabetes and renal insufficiency in a Spanish multi-center cohort [10]. In addition, the negative prognostic impact of comorbidities as assessed by the CCI was alleviated if a spleen reduction could be achieved following the administration of ruxolitinib [14]. Within this context, the importance of the JAK/STAT pathway as a therapeutic target for cardiovascular disease in myeloproliferative neoplasms has recently been reviewed [36]. 

### Limitations

In the interpretation of our data, one must consider several limitations. As a monocentric and retrospective chart, our study was subject to a selection bias, especially as not all data were available for all patients. One additional important aspect was the limited number of patients, due to which we had to combine PMF and MF post-ET/PV. However, for the latter patients, using the Myelofibrosis Secondary to PV and ET-Prognostic Model (MYSEC-PM) would have been more appropriate [37]. The small patient number precluded a systematic comparison of the MDS-CI with other comorbidity indices. In addition, it was not possible to adjust for potentially confounding factors like age or treatment with ruxolitinib. Furthermore, conventional cytogenetics were only available for a minority of patients, preventing the assessment of the MDS-CI within the context of MF-specific scoring systems that take into account information about the mutational status and chromosomal aberrations like the MIPSS-70/Plus V2.0 [38].

## 5. Conclusions

Our observations on the prognostic impact of comorbidities as determined by the MDS-CI in MF confirmed the importance of comorbidities, especially cardiac disease and solid tumors, for the course of the disease and overall survival in MF. In order to identify the best suitable tool to assess the impact of comorbidities, analyses from larger patient populations, e.g., from prospective and multi-institutional registries, would be desirable. This would allow a direct comparison of the MDS-CI and other scores like the HCT-CI or the ACE-27. 

## Figures and Tables

**Figure 1 cancers-15-04698-f001:**
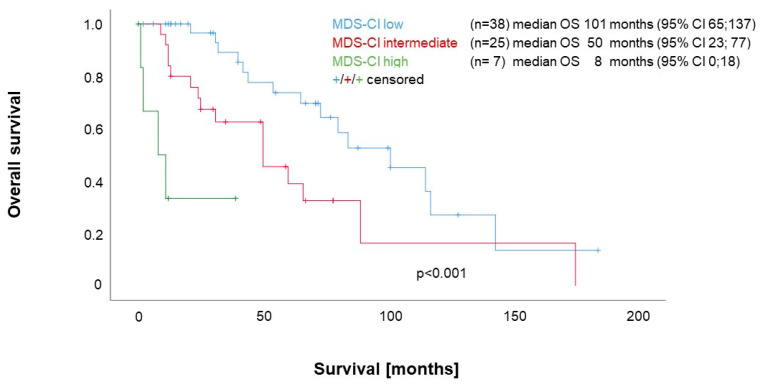
Impact of the MDS-CI on survival in MF patients.

**Figure 2 cancers-15-04698-f002:**
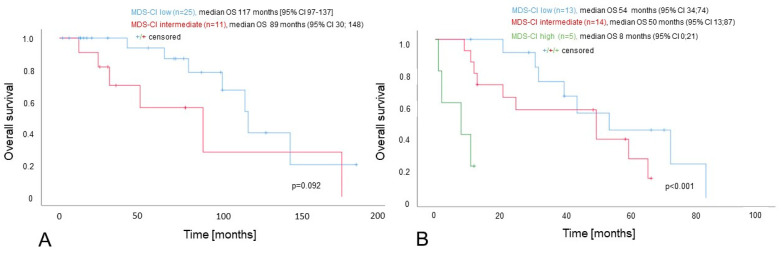
Impact of the MDS-CI on survival of MF patients stratified according to DIPSS. Panel (**A**) is a combined analysis of 36 patients with either DIPSS low- (n = 8) or intermediate (n = 28)-risk (“DIPSSdich^low^”). Panel (**B**) is a combined analysis of 32 patients with either DIPSS intermediate-2- (n = 24) or high (n = 8)-risk (“DIPSSdich^high^”).

**Figure 3 cancers-15-04698-f003:**
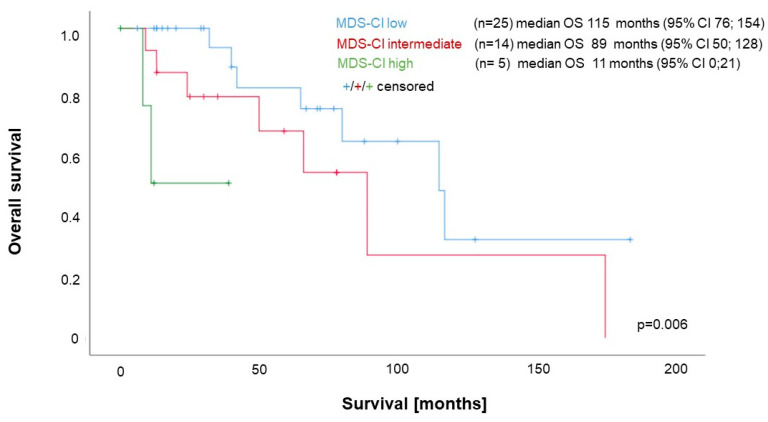
Impact of the MDS-CI on survival of MF patients stratified according to MIPSS70. Combined analysis of 44 patients with MIPSS70 low (n = 2) or intermediate (n = 42) (“MIPSS70dich^low/int^”).

**Table 1 cancers-15-04698-t001:** Characteristics of the MF cohort, including PMF (51/70) and MF post-ET/PV (19/70). Data are shown for the whole population and according to the presence or absence of comorbidities as defined by the MDS-CI (MDS-CI 0 versus MDS-CI ≥ 1).

	Whole Population	MDS-CI 0	MDS-C1 ≥1	*p*-Value
n	70	38	32	
Age (years), median (IQR)	73 (63–78)	70 (59–76)	77 (67–80)	0.005
Female n (%)	33/70 (47.1)	18/38 (47.4)	15/32 (46.9)	1.00
Bone marrow fibrosis grade 2, n (%)	49/70 (70)	24/38 (63.2)	25/32 (78.1)	
Bone marrow fibrosis grade 3, n (%)	21/70 (30)	14/38 (36.8)	7/32 (21.9)	0.200
Hemoglobin (g/L), median (IQR)	110 (88–123)	114 (99–124)	99 (83–121)	0.128
Platelet count (×10^9^/L), median (IQR)	411 (199–683)	481 (197–697)	391 (222–648)	0.700
Leukocytes (×10^9^/L), median (IQR)	9.3 (6.5–16.0)	7.7 (6.4–13.4)	11.4 (7.0–21.0)	0.067
Neutrophils (×10^9^/L), median (IQR)	6.6 (3.9–12.8)	6.1 (3.8–10.5)	7.7 (4.7–15.1)	0.166
Monocytes (×10^9^/L), median (IQR)	0.57 (0.35–0.84)	0.64 (0.41–0.83)	0.46 (0.27–1.09)	0.489
Lymphocytes (×10^9^/L), median (IQR)	1.5 (1.0–2.2)	1.5 (1.0–2.0)	1.5 (1.0–2.3)	0.781
Blasts PB (%), Median (IQR)	0 (0–1)	0 (0–1)	0 (0–1)	0.075
Constitutional symptoms, n (%)	33/70 (47.1)	14/38 (36.8)	19/32 (59.4)	0.092
LDH available (U/L), median (IQR)	62/70 530 (355–686)	33/38 525 (365–659)	29/32 554 (327–789)	0.672
CRP available (mg/L), median (IQR)	65/70 5 (2–12)	33/38 3 (1–7)	32/32 9 (5–29)	< 0.001
Ferritin available (μg/L), median (IQR)	49/70 151 (69–275)	24/38 122 (55–176)	25/32 210 (116–396)	0.009
Albumin available (g/L), median (IQR)	55/70 39.9 (37.0–42.3)	28/38 41.9 (37.9–42.9)	27/32 38.2 (36.2–42.1)	0.056
Need for transfusion, n (%)	23/70 (32.9)	11/38 (28.9)	12/32 (37.5)	0.610
Splenomegaly (clinical or imaging)	56/70 (80)	28/38 (73.7)	28/32 (87.5)	0.231
BMI available (kg/m^2^), median (IQR)	65/70 24.5 (21.2–28.0)	33/38 23.2 (21.0–28.0)	32/32 25.0 (21.2–28.1)	0.512

LDH: lactate dehydrogenase; CRP: C-reactive protein; BMI: body mass index.

**Table 2 cancers-15-04698-t002:** Multivariate Cox regression models, including the MDS-CI and “DIPSSdich” (Multivariate 1, n = 70) or “MIPSS70dich” (Multivariate 2, n = 56) as categorical variables.

	Multivariate 1			Multivariate 2		
	HR	95% CI	*p*-Value	HR	95% CI	*p*-Value
MDS-CI *						
Intermediate	1.97	0.96; 4.03	0.065	1.63	0.65; 4.06	0.2961
High	14.64	4.42; 48.48	<0.001	19.65	4.71; 81.95	<0.001
DIPSSdich **	6.08	2.35; 15.71	0.0002			
MIPSS70dich ***				4.53	1.64; 12.53	0.0036

* Reference low-risk; ** reference low/intermediate-1-risk; *** reference low/intermediate-risk.

**Table 3 cancers-15-04698-t003:** Likelihood ratio tests and C-indices for the different models (DIPSSdich and MIPSS70dich with and without the MDS-CI).

Model	N	LL	df	AIC	BIC	LR Test *p*-Value	C-Index
DIPSSdich	70	−105.0	1	211.9	214.2		0.6999
DIPSSdich and MDS-CI	70	−98.7	3			0.0018	0.7814
MIPSS70dich	56	−73.4	1	148.7	150.8		0.6515
MIPSS70dich and MDS-CI	56	−67.0	1	139.9	146.0	0.0017	0.7770

LL: log of likelihood; df: degrees of freedom.

## Data Availability

The data presented in this study are available on request from the corresponding author.

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
