# Peer review of "Prognostic Significance of the Myelodysplastic Syndrome-Specific Comorbidity Index (MDS-CI) in Patients with Myelofibrosis: A Retrospective Study"

_cancers, 2023, doi:10.3390/cancers15194698_

Round 1

Reviewer 1 Report

This paper aims to analyze the impact of MDS-CI  (MDS comorbidity index) applied in MF setting. 

The results are related to a small case series, in which MF patients who were candidates for transplantation , therefore younger and/or with a poor prognosis (IPSS INT-1 with poor mutations and IPSS INT-2/High risk), were excluded, determining a selection bias. Primary and secondary MF were considered together, so the statistical analysis of the differences between high and low MDS-CI in MF patients for to concern age, laboratory variables, degree of fibrosis, extent of splenomegaly is difficult to interpret. The patients with higher MDS-CI have worse prognosis, but how they were treated is not detailed. Considering the most presence of comorbidities in high MDS-CI group (heart + tumors), it could be that these patients have not been adequately treated, according to IPSS risk. Consequently the conclusions of the paper are a bit weak and not very applicable to clinical practice. Considering the high IF of Cancers, I suggest submitting to another journal with a lower IF, subject to revision with the suggestions indicated.

a modest English revision is needed

Reviewer 2 Report

The paper “Prognostic Significance of the Myelodysplastic Syndrome-specific Comorbidity Index (MDS-CI) in patients with myelofibrosis — A Retrospective Study by Koster et al.  deals with a very important issue. This is a retrospective work with a very clinical approach that certainly makes a useful contribution to the scientific community.

Myelodysplastic syndromes (MDSs) are a group of myeloid clonal hemopathies with a relatively heterogeneous spectrum of presentation, characterized by varying degrees of cytopenias and a predisposition to acute myeloid leukemia. Several scoring systems, such as IPSS and its revised version IPSS-R, DIPSS have been developed to stratify MDS. However, these scoring systems depend mainly on the percentage of the myeloblasts, karyotypic abnormalities, and the number of peripheral blood cytopenias. Increasing evidence suggests that MF is an adverse factor in MDS, which is associated with early bone marrow failure and shorter survival, although this risk feature is not captured by IPSS-R. MF can present as primary myelofbrosis (PMF), or accompanied by / arise from a pre-existing diagnosis of MDS, polycythemia vera or essential thrombocythemia and so on. The most important difference between PMF and MDS with MF is the apparent multilineage lineage dysplasia observed in the latter. What’s more, there are less heteromorphic and broken erythrocytes and there is no obvious hepatosplenomegaly in MDS with MF. As part of the disease process, patients with MF often present with disease-related comorbidities, such as progressive anemia causing chronic fatigue and weakness, splenomegaly and/or hepatomegaly due to extramedullary hematopoiesis, and constitutional symptoms such as night sweats, weight loss, pruritus, low-grade fever, and bone pain.

There is some debate among clinicians about when the issue of “comorbidities” should arise during a case presentation. Some believe it should be mentioned first, while others think it occupies a less urgent priority and should only be mentioned later. Nevertheless, everyone agrees that a case presentation is incomplete without the mention of comorbidities, given that they can heavily impact treatment strategy. As patients get older, the list of comorbidities tends to get longer. This is especially the case if there is an exacerbating lifestyle behavior, such as smoking or excessive alcohol consumption. As a result, a physician is never just treating 1 medical condition; it is more accurate to say that a physician is treating the most pressing of several medical conditions, all of which require adequate care. Like it or not, physicians need to contend with all the issues, the comorbidities, that a patient is faced with. A more enlightened approach is to see comorbidities as possibly mutually exacerbating, diverting the immune system into various individual battles that weaken the whole. Mismanaged, a comorbidity can kill a patient more quickly than their immediate presenting complaint. In no other circumstances is the impact of comorbidities as marked as when a patient’s immune system is undergoing an immense stress test, such as in the case of a cancer diagnosis and its subsequent treatment. Where we are at with cancer treatment today is that even a person with an otherwise clean bill of health is likely to undergo significant health challenges over the course of their cancer treatment. 

Previous studies have shown that the coexistence of cancer and other chronic conditions has substantial implications for treatment decisions and outcomes for both neoplasms and chronic disease, as reported by Garcia-Fortes. Coming from the same study, the influence of comorbidities on long-term outcomes among patients with myelofibrosis. They conducted an observational prospective study based on data from a Spanish multicenter myelofibrosis registry. Recruited participants must be adult patients diagnosed with myelofibrosis according to World Health Organization standards (n=668). The research team took note of the comorbidities present among recruited participants at the time of diagnosis and performed a risk stratification based on that information. The main effect of these comorbidities is that it interferes with a patient’s fitness to undergo specific myelofibrosis therapies that may improve outcomes. This problem is confounded by the fact that most clinical studies on myelofibrosis treatment exclude patients with significant organ dysfunction, meaning that a gap in the literature exists on how best to treat myelofibrosis patients with significant comorbidities. It is, therefore, imperative that patients with myelofibrosis are assessed holistically before making treatment decisions and determining the most likely course of the disease. Adding patient-specific comorbidities improved the prognostic effect of risk prediction models for patients with primary myelofibrosis (PMF) or secondary myelofibrosis (sMF), according to findings from an assessment of data collected in Vanderbilt’s Synthetic Derivative and BioVU Biobank comprehensive electronic health record (EHR). Discrimination power was significantly higher using the extended Dynamic International Prognostic Scoring System (DIPSS) model that incorporated renal failure/dysfunction, intracranial hemorrhage, invasive fungal infection, and chronic encephalopathy (C-index, 0.81; 95% CI, 0.78-0.84) compared with the original DIPSS model (C-index, 0.73; 95% CI, 0.70-0.77). In this above paper Sokoki reports that in aggregate, their findings suggest that a more objective measurement of patient-specific comorbidities is needed to best individualize therapy in this highly comorbid patient population.

In this context, this article under review may contribute to a better management of MF, as the proposed MDS-CI index offers additional prognostic information after inclusion in a multivariate model together with dichotomised DIPSS or MIPSS70, as reported and demonstrated in the article. Their results support the use of Ruxolitinib for the treatment of patients at low risk of MF when comorbidities are established.

The paper has two relevant limitations due to the low number of enrolled patients and to the almost missing data on cytogenetics that for this type of study is a key clinical parameter. Being a retrospective study, however, it could not have been otherwise. On the other hand, the journal accepts that these two critical issues are reported in the dedicated Limitations section. For the rest, the study is well conducted, the experimental design is clear and the results confirm the hypothesis of a tool that can better stratify patients with MF for a more targeted therapy.

The paper in this version can be accepted for publication on Cancers.

English is almost fine

Reviewer 3 Report

The authors showed that in myelofibrosis, comorbidities add prognostic information independently of the International Dynamic Predictive Scoring System (DIPSS). The Myelodysplastic Syndrome-Specific Comorbidity Index (MDS-CI) is a simple tool for assessing CM because it can be calculated after a thorough history and physical examination, a small standard biochemical panel including creatinine and liver enzymes, and a limited set of functional measures diagnostics. In doing so, MDS-CI added predictive information after inclusion as a categorical variable in the multivariate model along with dichotomized DIPSS or dichotomized MIPSS70.

In general, I liked the article. There are some minor design flaws. The small sample size suggests that the study is rather preliminary and requires additional verification, which the authors cited as one of the limitations of the study. Is it worth it to include groups in Figure 2a and Figure 3b, which include 2 people. I would recommend that the authors remove these groups from consideration and confine themselves to comparing only more representative samples.

Reviewer 4 Report

In this manuscript, the authors applied the Myelodysplastic Syndrome-specific comorbidity index (MDS-CI) in MF. They suggested that the MDS-CI represents a valuable tool to identify MF patients with an increased vulnerability due to comorbidities. However, this paper is unacceptable for publication in its current form. The specific comments are listed below:

1.     Please specify the reasons for studying the impact of the MDS-CI in the context of the DIPSS and MIPSS70 in the article.

2.     The number of samples is limited, whether the data can be enriched by combining multiple centers.

3.     On page 5, part 3.3, the author said, “As shown in Figure 1, median overall survival (OS) was significantly influenced by the presence of comorbidities classified according to the MDS-CI at the time-point of diagnosis.” However, Figure 1 illustrates the survival of all patients instead of patients with comorbidities.

4.     The reason should be explained why MDS-CI classification in the DIPSSdichlow group had no significance.

5.     Is it possible for the authors to analyze whether MDS-CI can act as an independent prognostic factor in combination with known prognostic factors?

Minor editing of English language required

Round 2

Reviewer 1 Report

The authors tried to answer most of the questions posed, improving the paper and adding material in the supplements.I believe you can proceed with the publication of this paper in agreement of the opinion of other reviewers. 

not applicable

Reviewer 3 Report

I have no further comments on the article. However, it would be more convenient if the authors highlighted the changes in the text that they made in accordance with the comments of mine and other reviewers.

Reviewer 4 Report

In this manuscript, the authors applied the Myelodysplastic Syndrome-specific comorbidity index (MDS-CI) in MF. They suggested that the MDS-CI represents a valuable tool to identify MF patients with an increased vulnerability due to comorbidities. This article is innovative and the data processing is reasonable which has certain clinical application prospect and value. Except for a few sentences structure is relatively monotonous, this article can be accepted for publication.

Minor editing of English language required.